

# Persian sentiment analysis of an online store independent of pre-processing using convolutional neural network with fastText embeddings

Sajjad Shumaly[1], Mohsen Yazdinejad[2] and Yanhui Guo[3]

[1] Industrial Engineering, Sharif University of Technology, Tehran, Iran
[2] Computer Engineering, Isfahan University, Isfahan, Iran
[3] Computer Science, University of Illinois at Springfield, Springfield, IL, USA

Corresponding author
Yanhui Guo, yguo56@uis.edu

## ABSTRACT

Sentiment analysis plays a key role in companies, especially stores, and increasing the accuracy in determining customers' opinions about products assists to maintain their competitive conditions. We intend to analyze the users' opinions on the website of the most immense online store in Iran; Digikala. However, the Persian language is unstructured which makes the pre-processing stage very difficult and it is the main problem of sentiment analysis in Persian. What exacerbates this problem is the lack of available libraries for Persian pre-processing, while most libraries focus on English. To tackle this, approximately 3 million reviews were gathered in Persian from the Digikala website using web-mining techniques, and the fastText method was used to create a word embedding. It was assumed that this would dramatically cut down on the need for text pre-processing through the skip-gram method considering the position of the words in the sentence and the words' relations to each other. Another word embedding has been created using the TF-IDF in parallel with fastText to compare their performance. In addition, the results of the Convolutional Neural Network (CNN), BiLSTM, Logistic Regression, and Naïve Bayes models have been compared. As a significant result, we obtained 0.996 AUC and 0.956 $F$-score using fastText and CNN. In this article, not only has it been demonstrated to what extent it is possible to be independent of pre-processing but also the accuracy obtained is better than other researches done in Persian. Avoiding complex text preprocessing is also important for other languages since most text preprocessing algorithms have been developed for English and cannot be used for other languages. The created word embedding due to its high accuracy and independence of pre-processing has other applications in Persian besides sentiment analysis.

# INTRODUCTION

With the advancement of technology and the spread of the Internet, proper conditions have been provided for the activities of online stores. Due to some advantages such as high

variety, delivery speed, and time savings, customers of this type of store are constantly increasing (*Liang & Wang, 2019*). When buying from online stores, due to the gap between the buyer and the product, there may be some problems such as poor quality of products, inadequate after-sales service, or inconsistency between product descriptions and performance (*Ji, Zhang & Wang, 2019*). One of the viable solutions to overcome the problems is to use the opinion of users who have already purchased the product.

In the past, if people needed to know the other's opinion, they would ask questions of family, friends, or relatives. Similarly, companies and stores used surveys to find out the opinions of people or customers. But today, if people require to buy or companies and stores need to know the opinions of customers to provide better services and products, they can easily refer to people's comments and discussions on online store websites or forums. Therefore, online reviews are important sources of information about the quality of goods that play a key role in customer awareness of products (*Li, Wu & Mai, 2019*). Online reviews enable the customer to have a comprehensive view of the products and their alternatives before making a purchase; thus, it has a significant impact on the extent of product sales (*Hu, Liu & Zhang, 2008*). As a matter of fact, the immediate response of stores to their customers' complaints is essential in maintaining their competitive position. But analyzing these reviews manually is quite time-consuming and costly. Also, automatic comment analysis has some obstacles; problems such as using sentences with incorrect grammar, using slang terms, and not following the correct punctuation are an integral part of making text analysis difficult (*Irfan et al., 2015*). When it comes to resolving these problems, sentiment analysis techniques play an essential role. These techniques automatically estimate customer sentiment into positive, negative, and even neutral classes. Therefore, sentiment analysis for online stores is highly valued because it can extract users' sense of goods and help to make decisions to increase customer satisfaction and product sales. Sentiment analysis can be considered as a type of content analysis that specifically seeks to determine the emotional tone of the text (*Oscar et al., 2017*). This is done based on the emotional evidence between words and phrases (*Tausczik & Pennebaker, 2010*).

In this article, we are seeking to analyze the feelings of customer reviews on the website of the largest and well-known online store in Iran (Digikala). At first, lingual problems were taken into account as a significant challenge. There are several problems in Persian text pre-processing such as using slang, using letters of other languages especially Arabic, lack of a clear boundary between phrases. To tackle the problems, we employed fastText and skip-gram because we wanted to examine whether the utilize of the methods capable of reducing the need for data pre-processing and make language processing easier. In the following, we will inspect this assumption and compare the obtained results with other algorithms and other reports. Another severe limitation was that the deep learning models required an immense dataset, but most of the available datasets in Persian are small to such an extent that they cannot be employed in deep models. Thus, a rich and immense dataset had to be extracted from the Digikala website which was conducted

by web-mining methods. It should be noted that this article seeks to achieve the following goals:

- Investigating the reduction of the need for text pre-processing by implementing methods such as fastText and skip-gram, either in Persian language processing or others;
- Gathering comprehensive customer reviews dataset based on various types of digital goods to create a general word embedding for a various range of works related to digital goods;
- Sentiment analysis of Digikala website's reviews with high accuracy even compared to other researches.

## RELATED WORKS

Sentiment analysis methods are divided into three general categories: Lexicon based, traditional Machine Learning, and Deep Learning models (*Yadav & Vishwakarma, 2020*). The first category is the sentiment analysis using a sentiment lexicon and it is an unsupervised method. In this case, emotional similarities of words and phrases are used and its accuracy is highly dependent on pre-learned weights (*Taboada et al., 2011*). This method collects a set of pre-compiled sentiment words, terms, phrases, and idioms with a specific thematic category such as opinion finder lexicon (*Wilson et al., 2005*) and ontologies (*Kontopoulos et al., 2013*).

The second category is based on machine learning methods which are divided into supervised and unsupervised categories. The accuracy of these methods is strongly influenced by the extracted features from the text. Supervised techniques such as Naïve Bayes, SVM, Maximum Entropy and Logistic Regression are the most common techniques in this field (*Ye, Zhang & Law, 2009*; *Montejo-Ráez et al., 2014*). However, unsupervised methods are suitable for situations where labeling for the dataset is impossible (*Paltoglou & Thelwall, 2012*).

Deep learning has grown and been used in many areas in the last decade, for example in the field of object recognition (*Ghoreyshi, AkhavanPour & Bossaghzadeh, 2020*; *Ali et al., 2020*), speech recognition (*Deng, Hinton & Kingsbury, 2013*; *Li, Baucom & Georgiou, 2020*), anomaly detection (*Zhao et al., 2018*), feature extraction (*Lin, Nie & Ma, 2017*; *Rajaraman et al., 2018*), auto-encoding (*Pu et al., 2016*). Also, in cases where deep learning along with machine learning has been used for text analysis and sentiment analysis, good results have been obtained (*Tang, Qin & Liu, 2015*; *Severyn & Moschitti, 2015*). The main difference between sentiment analysis by deep learning and other methods is how to extract the features. To be specific, one of the advantages of deep learning models is that there is no need for user intervention in feature extraction, which of course requires a large amount of data to perform the feature extraction operation. Recurrent Neural Network (RNN), Convolutional Neural Network (CNN), Long Short-Term Memory (LSTM) and Gated Recurrent Unit are the most common models of deep learning in sentiment analysis (*Zhang, Wang & Liu, 2018*).

The most basic and widely used CNN model for sentiment analysis at the sentence level is the one presented by *Kim (2014)*. Then, *Zhang & Wallace (2015)* proposed a special single-layer CNN architecture by examining improvements made by changing the model configuration. Many developments have been made to improve the performance of CNN-based sentiment analysis models. In this regard, an example of CNN combined with fuzzy logic called the Fuzzy Convolutional Neural Network (*Nguyen, Kavuri & Lee, 2018*) is noteworthy. The use of CNN in natural language processing is now a common topic and much research is being done in this area (*Wehrmann et al., 2017*; *Gan et al., 2020*; *Arora & Kansal, 2019*).

Deep neural networks are difficult to train because they often suffer from the problem of vanishing gradients. LSTM architecture was introduced (*Hochreiter & Schmidhuber, 1997*) to overcome this shortcoming to learn long-term dependencies. After the original work, the LSTM has experienced several improvements such as adding forget gate (*Gers, 1999*). A neural network architecture could not be so great adopted into practice without strong theoretical support, therefore, a widespread review considering the several LSTM variants and their performances relative to the so-called vanilla LSTM model was conducted by *Greff et al. (2017)*. The vanilla LSTM model is interpreted as the primary LSTM block with the addition of the forget-gate and peephole connections. Also, to overcome some limitations in conventional RNN models, bidirectional RNN (BRNN) models were proposed. Using this model's structure, both future and past situations of sequential inputs in a time frame are evaluated without delay (*Schuster & Paliwal, 1997*). By combining the ideas of BRNN and LSTM it is possible to achieve Bidirectional LSTM (BiLSTM) which has better performance than LSTM in classification processes. With the development of LSTM in recent years, it has been used in projects such as Google Translate and Amazon Alexa (*Wu et al., 2016*; *Vogels, 2016*) in natural language processing.

## MATERIALS AND METHODS

All the taken steps, methods, codes, and results that are presented below, along with a part of the extracted dataset are fully accessible on the repository (*Yazdinejad & Shumaly, 2020*).

### Dataset

Having access to a large dataset with richness and content integrity is indispensable to train a deep model. Most available datasets to train a deep model and sentiment analysis are in English. To collect a rich dataset, web-mining methods were used and the reviews on the Digikala website were extracted which were in Persian. Posted reviews by buyers express their level of satisfaction with their purchase and product features. After submitting their reviews, buyers could choose between the "I suggest" and "I do not suggest" options. These two options were extracted and used in the model as labels for the problem of sentiment analysis. Our goal was to analyze the opinions of users of the Digikala website, so we extracted the data of the section related to digital goods using web-mining libraries such as the Beautiful Soup (*Richardson, 2020*). Beautiful Soup is a Python package to parse XML and HTML documents and it is useful for web scraping (*Hajba, 2018*).

## Pre-processing

One of the first steps in natural language processing problems has always been data pre-processing. At this stage, the texts need to be cleaned and prepared to begin the analysis. In Persian, this stage is even more difficult and important because it has its complexities. This field has attracted many researchers in the last decade, therefore, libraries and algorithms in the field of pre-processing in Persian have been developed (*Mohtaj et al., 2018*; *Nourian, 2013*) which have become more complete and better over time. However, these algorithms cannot work as well as similar algorithms in English and need further development. We are seeking a way to achieve an accurate result by avoiding the complications of data pre-processing steps in Persian. Regular expressions are used for data pre-processing in all of the following steps using the "re" library (*Rachum, 2020*) in python. Pre-developed libraries for the Persian language have not been used to perform data pre-processing steps and we assume that the use of fastText and skip-gram in creating word embedding reduces the need for complex pre-processing.

## Normalization

In Persian, some letters are not unique and may have different alternatives to other languages such as Arabic. For example, the letter "ی" is Persian, but the letter "ي" is Arabic, and these two letters will likely be used interchangeably. This causes the created words to be considered as two different words. In this way, they may be considered separately in the calculations and a vector can be drawn for each with its characteristics. To solve this issue, it is necessary to use the standard form for all available texts.

## Tokenization

Tokenization is a stage in which attempts are made to divide sentences into meaningful words and phrases that can be considered as a suitable input for the next steps. The main challenge of the Persian language at this stage is that sometimes there are no clear boundaries between phrases as a result of three different modes of spacing in Persian. In other words, phrases in Persian can be without space, with half-space, or with space, which is often mistakenly used instead of each other. For instance, the words "نرم افزار" and "نرمافزار", which both mean software, are written with both space and half-space forms. If the wrong form is used, the phrase border will be mistakenly recognized as two separate words "نرم" and "افزار". Vice versa, phrases that consist of several words can be considered as one word due to a mistake in using space. For example, the word "از مسیر دیگر", which means "from another path", maybe written as "ازمسیردیگر" without any spaces. These kinds of mistakes blur the line between phrases and words and make it difficult to pre-process.

## Stemming

The stemming process seeks to remove part of the word in such a way that the root of the word is determined (*Willett, 2006*). The root of the word does not necessarily mean the dictionary root of the word and is acceptable in cases where it can improve the performance of the model. For example, we can refer to the phrase "رنگهایشان".
In this phrase, "رنگ" means color, and "ها" is used to represent plural and "یشان" for

determination of ownership. A significant number of stemming algorithms use the following rule (*Mohtaj et al., 2018*):

(possessive suffix)(plural suffix)(other suffixes)(stem)(prefixes)

Stemming is a rule-based process that is usually done by removing suffixes and prefixes. Consequently, it cannot be used in other languages and each language requires its algorithms. Stemming algorithms are being developed in Persian but due to the complexity of the Persian language, their performance needs to be improved.

## Pseudo labeling

In classification problems, it is a common problem that a large number of samples do not have labels and therefore cannot be used in model training. Techniques such as pseudo-labeling can be used to overcome this problem and determine the labels of some samples (*Lee, 2013*). The first step in pseudo-labeling is to develop a model based on labeled samples in the dataset that is in charge of determining the label of unlabeled samples. Only labels that have been estimated with high confidence are accepted. In the next step, another model is developed using training data along with the new labeled data which can be used to predict test data with higher accuracy. In this way, 104.8 thousand Negative Feedback reviews and 30.5 thousand Positive Feedback reviews were labeled and could be used in the dataset for subsequent analysis. As will be shown in the results section, this method had a significant impact on improving accuracy.

## Data balancing

Unequal distribution of data in different classes in a classification problem leads to data imbalance. The class with the most data is called the majority class, and the class with the least data is called the minority class. In these cases, the models tend to ignore the minority class and predict in favor of the majority class. Many machine learning models, such as Support Vector Machine, Naïve Bayes, Decision Tree and Artificial Neural Network cannot have good results in this situation (*Díez-Pastor et al., 2015*; *Vorraboot et al., 2015*). In general, data balancing solutions can be divided into two categories; over-sampling and under-sampling. The goal of both solutions is to approximate the number of data distributed in the minority and majority classes. In over-sampling, this is done by increasing the amount of data in the minority class, and in under-sampling by reducing the amount of data in the majority class. In the present problem, we used the random oversampling method to balance the dataset.

## Feature extraction
### fastText

Neural network-based methods have become very popular in the field of natural language processing due to their accuracy. However, most of these methods are slow to analyze large datasets and they need to receive word embedding to analyze texts. For this reason, a method called fastText has been proposed (*Joulin et al., 2016*). fastText is an efficient, fast, and open-source model that Facebook has recently released. In fastText, a set of tricks has been used to improve the processing speed and performance of the model, one of

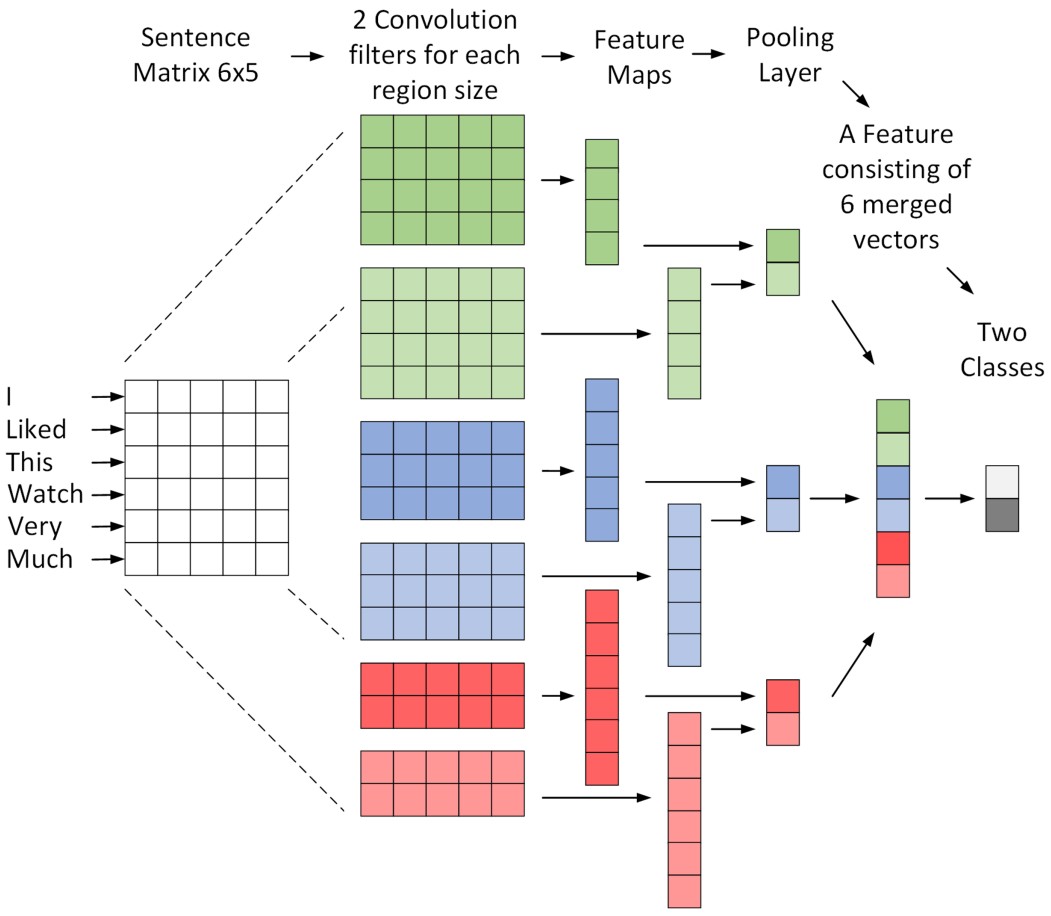

**Figure 1 A convolutional network architecture to sentiment classification.**

which is skip-gram. Data sparsity has always been one of the biggest problems in natural language analysis. In other words, the main problem of modern language processing is that language is a system of rare events, so varied and complex, that we can never model all possibilities (*Preethi Krishna & Sharada, 2020*). Therefore, skip-gram allows some words to be skipped and non-adjacent words to be examined together. *Mikolov et al. (2013)* found the skip-gram model to be superior to the bag-of-word model in a semantic-syntactic analogy task. Skip-gram is popular, easy to implement, and it is proven and reliable (*Gurunath & Prashanth, 2019*). Accordingly, in this article, word embeddings have been provided using fastText and skip-gram to investigate the reduction of language processing dependence on data-preprocessing.

## Sentiment analysis model

### Convolution neural network

Using CNN has shown high accurate results based on studies in English texts (*Nedjah, Santos & De Macedo Mourelle, 2019*). This model can receive and analyze word embedding as input instead of images, which are also effective in this area (*Kim, 2014*). Each row of the input matrix represents a word. Figure 1 shows the architecture of a CNN model

**Table 1 The CNN model structure.**

| Layer (type) | Output shape | Number of parameters |
|---|---|---|
| embedding_2 (Embedding) | (None, 400, 100) | 11,147,700 |
| dropout_8 (Dropout) | (None, 400, 100) | 0 |
| conv1d (Conv1D) | (None, 400, 128) | 38,528 |
| global_max_pooling1d (Global) | (None, 128) | 0 |
| dense_6 (Dense) | (None, 64) | 8,256 |
| dropout_9 (Dropout) | (None, 64) | 0 |
| dense_7 (Dense) | (None, 16) | 1,040 |
| dropout_10 (Dropout) | (None, 16) | 0 |
| dense_8 (Dense) | (None, 1) | 17 |
| Total parameters: | | 11,195,541 |

used for the NLP classification problem (*Zhang & Wallace, 2015*). This figure shows how a CNN model treats a 6-word sentence. The matrix formed for this sentence is analyzed by six different convolution filters and converted to maps of attributes with dimensions of $1 \times 4$, $1 \times 5$ and $1 \times 6$. Finally, the pooling operation is performed on the maps and their outputs are merged to create a unique vector that can be used as input for the SoftMax layer to determine the class. The CNN model used in this article is based on the mentioned model and its architecture is shown in Table 1.

### Bidirectional long short-term memory

Another deep model used to solve the problem is BiLSTM. The LSTM model can decide which information is useful and should be preserved and which information can be deleted based on the dataset it has trained with. The LSTM has been widely used in NLP such as long document categorization and sentiment analysis (*Rao et al., 2018*). Figure 2 is a demonstration of an LSTM cell used in this article, which has an input layer, an output layer and a forget layer (*Gers, 1999*). Based on the figure, the LSTM cell mathematically expressed as follows:

$$f_t = \sigma(W_{fh}h_{t-1} + W_{fx}x_t + b_f) \tag{1}$$

$$i_t = \sigma(W_{ih}h_{t-1} + W_{ix}x_t + b_i \tag{2}$$

$$\tilde{c}_t = \tanh(W_{\tilde{c}h}h_{t-1} + W_{\tilde{c}x}x_t + b_{\tilde{c}}) \tag{3}$$

$$c_t = f_t.c_{t-1} + i_t.\tilde{c}_t \tag{4}$$

$$o_t = \sigma(W_{oh}h_{t-1} + W_{ox}x_t + b_o) \tag{5}$$

$$h_t = o_t \cdot \tanh(c_t) \tag{6}$$

where $x_t$ denotes the input; $h_{t-1}$ ,and $h_t$ denote the output of the last LSTM unit and current output; $c_{t-1}$, and $c_t$ denote memory from the last LSTM unit and cell state; $f_t$ denotes forget gate value; $W_i$, $W_{\tilde{c}}$, and $W_o$ are the weights; $b$ is the bias; the operator '·' denotes the pointwise multiplication of two vectors. In LSTM, the input gate can decide what new information can be stored in the cell state, also the output gate can decide what

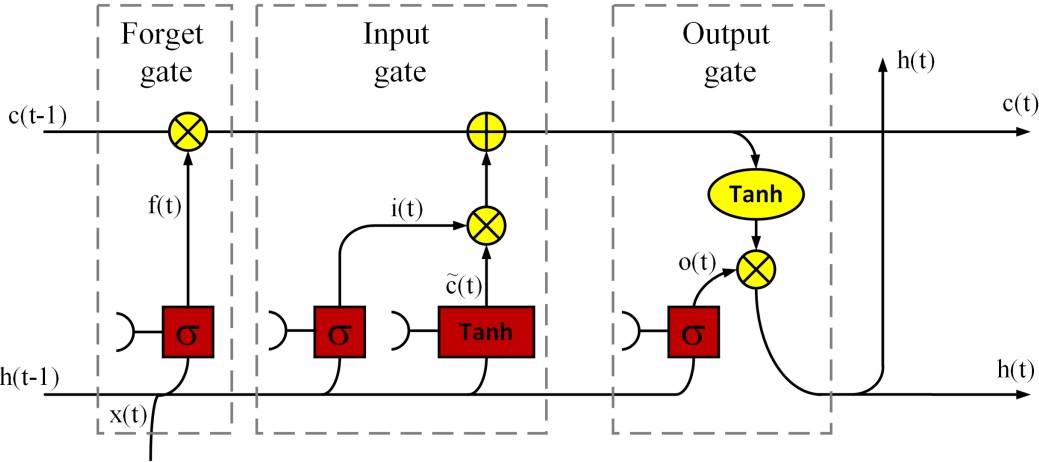

**Figure 2 A demonstration of an LSTM cell.**

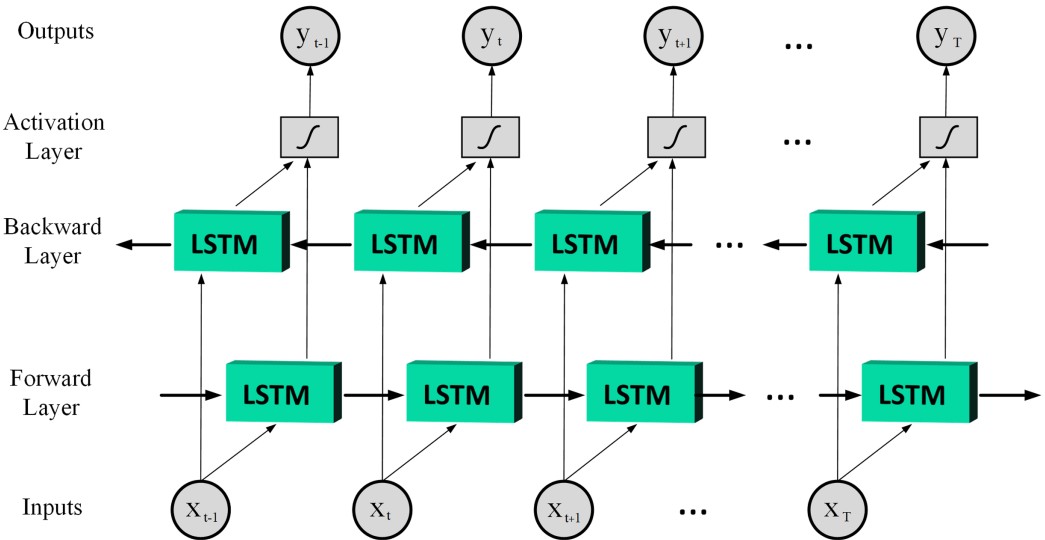

**Figure 3 A basic structure of the BiLSTM network.**

information can be output based on the cell state. By combining the ideas of BRNN and LSTM it is possible to achieve Bidirectional LSTM (BiLSTM) which has better performance than LSTM in classification processes especially in speech processing tasks (*Graves & Schmidhuber, 2005*). Therefore, this article uses the BiLSTM structure, and Fig. 3 is shown a basic structure of the BiLSTM network (*Yildirim, 2018*). The BiLSTM model used in this article architecture is shown in Table 2.

## Evaluation

Due to imbalanced data, indicators such as accuracy is not appropriate for this study. Because the developed model in the face of this type of data tends to ignore the minority class and can still be accurate. For this purpose, AUC and F-score indexes

**Table 2 The BiLSTM model structure.**

| Layer (type) | Output shape | Number of parameters |
|---|---|---|
| embedding_1 (Embedding) | (None, 400, 100) | 11,147,700 |
| dropout_4 (Dropout) | (None, 400, 100) | 0 |
| bidirectional_1 (Bidirection) | (None, 64) | 34,048 |
| dropout_5 (Dropout) | (None, 64) | 0 |
| dense_3 (Dense) | (None, 64) | 4,160 |
| dropout_6 (Dropout) | (None, 64) | 0 |
| dense_4 (Dense) | (None, 16) | 1,040 |
| dropout_7 (Dropout) | (None, 16) | 0 |
| dense_5 (Dense) | (None, 1) | 17 |
| Total parameters: | | 11,186,965 |

will be used, which are good choices for problems dealing with imbalanced data (*Sokolova, Japkowicz & Szpakowicz, 2006*). AUC indicates the area below the diagram in the ROC curve, and the ROC curve is a method for judging the performance of a two-class classifier (*Luo et al., 2020*). In the ROC curve, the vertical axis is the TPR (represents the true positive rate), Also, the horizontal axis is FPR (represents the false positive rate).

$$\text{FPR} = \frac{\text{fp}}{\text{tn} + \text{fp}} \qquad (7)$$

$$\text{TPR} = \frac{\text{fp}}{\text{tp} + \text{fn}} \qquad (8)$$

– TP: positive samples are classified as positive

– FN: positive samples are classified as negative

– TN: negative samples are classified as negative

– FP: negative samples are classified as positive

The *F*-score is the harmonic mean of precision and recall (*Velupillai et al., 2009*) and represents a weighted average of precision and recall (*Gacesa, Barlow & Long, 2016*). This index has wide applications in natural language processing (*Derczynski, 2016*), and like the AUC, it can be used in problems involved with imbalanced data.

$$\text{Precision} = \frac{\text{tp}}{\text{tp} + \text{fp}} \qquad (9)$$

$$\text{Recall} = \frac{\text{tp}}{\text{tp} + \text{fn}} \qquad (10)$$

$$F\text{-score} = 2 \times \frac{\text{Precision} \times \text{Recall}}{\text{Precision} + \text{Recall}} \qquad (11)$$

All the steps mentioned in the methodology section can be summarized in Fig. 4.

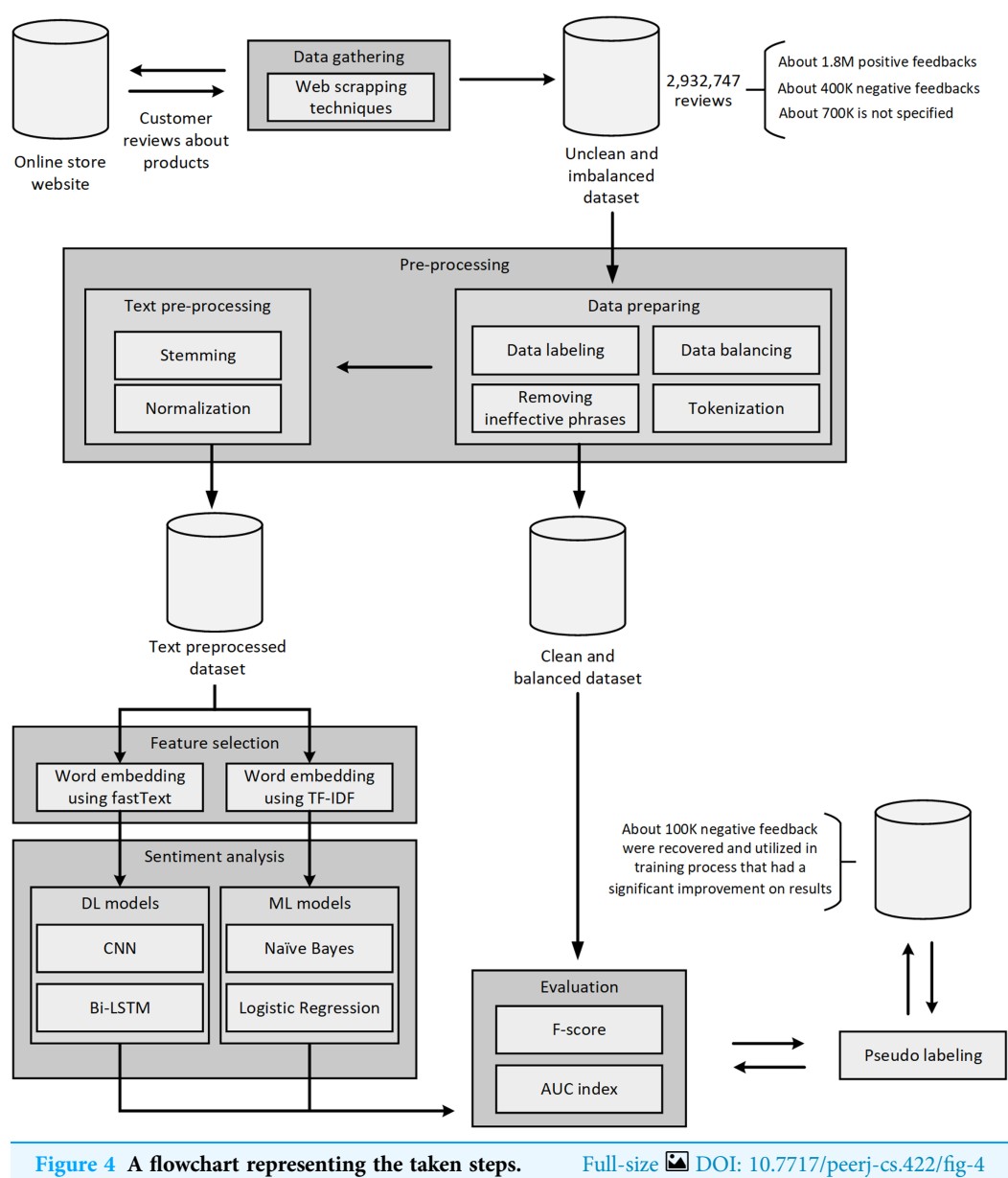

**Figure 4  A flowchart representing the taken steps.**

## RESULTS AND DISCUSSION

### Dataset

The digital goods' reviews of the Digikala website were extracted, which are a total of 2,932,747 reviews. Figure 5 shows the frequency of comments per category. Examining the comments of different product categories can increase the comprehensiveness of the model. To be specific, the words, phrases, and sentences are different in reviews of the products in the different categories, and considering various types of product categories will improve the generalization of the model in different situations. Table 3 shows the general structure of the collected dataset. In this table, the "Comment ID" column stores the unique values for each comment, the "Original Comment" column is the original of the

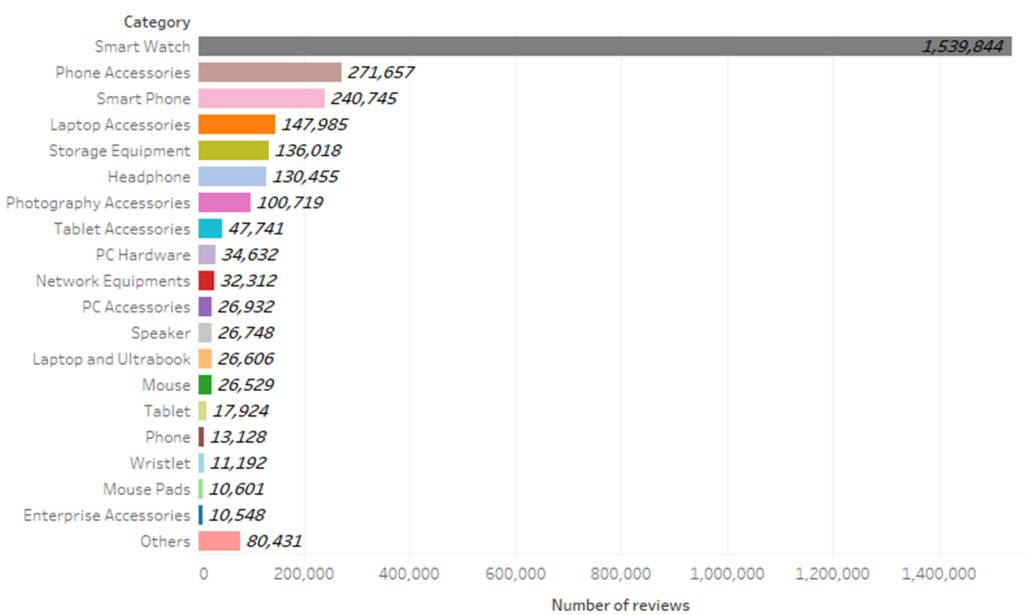

**Figure 5 Frequency of reviews per category.**

**Table 3 A sample of the collected dataset.**

| Comment ID | Original comment | Translated comment | Negative feedback | Positive feedback | Cat. Name |
|---|---|---|---|---|---|
| 0 | من کاملا با این محصول آشنا بودم و از خریدش مطمین بودم | I was completely familiar with this product and I was sure of buying it | 0 | 1 | Smart watch |
| 31 | به نسبت قیمتش عاااليه | Grrrreat for the price | 0 | 1 | Smart Watch |
| 84278 | چیز بدی نیست کار راه میندازه | Not a bad thing | 0 | 0 | Phone Accessories |
| 3083 | با توجه به معرفی پرچمدار جدید اچ تی سی u11 خرید این گوشی عاقلانه نیست. | Given the introduction of the new flagship HTC u11, buying this phone is not a wise choice | 1 | 0 | Smart Phone |
| 1503094 | رنگ مد نظر ارسال نشد | The requested color was not sent | 1 | 0 | Smart Watch |

comments written in Persian, the "Translated Comment" column is a translation of the "Original Comment" column into English. The "Translated Comment" column is used only to increase readability in the table and does not exist in the dataset. In the "Negative Feedback" column, if the value is 1, means that the user is not advised to buy the product, and in the "Positive Feedback" column, if the value is 1, it means the user is advised to buy the product, and the "Cat. Name" column represents the product category for which the comment was written.

The positive point of this website is that the buyers after submitting their comments can choose an option that states whether they generally recommend the product to others or not. Therefore, a significant number of extracted reviews are labeled. In other words, 308,122 of the reviews in the dataset do not recommend purchased items to others and the "Negative Feedback" column of these reviews in the dataset shows the number one. Likewise, 1,749,055 of the reviews in the dataset recommend the purchased items to

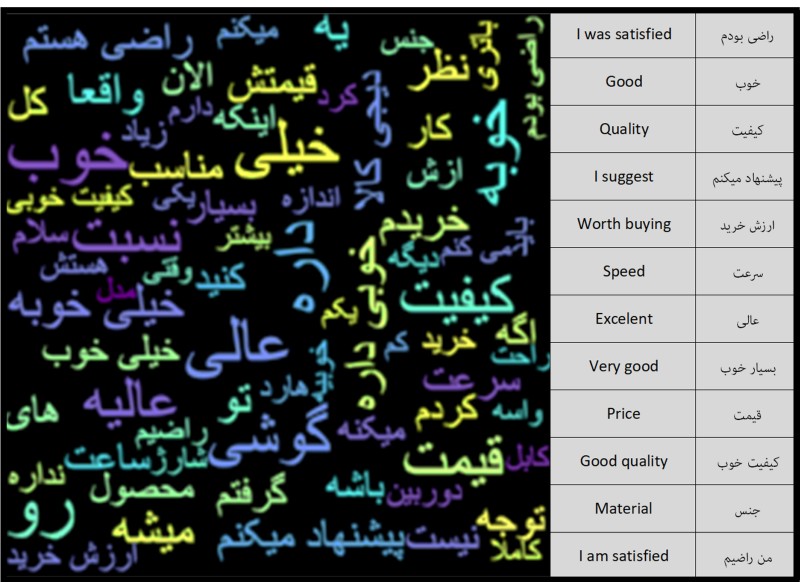

**Figure 6 Positive feedback class word cloud.**

others, and the "Positive Feedback" column of these comments in the dataset shows the number one. A significant part of the reviews is without labels and the reviews with labels are also imbalanced and these problems must be addressed in some ways.

## Pre-processing

During the initial review of the comments, the first correction that had to be made was the removal of escape sequences. An escape sequence is a series of two or more characters starting with a backslash and when used in a string, are not displayed as they are written. In most reviews, there were some escape sequences such as "\n" and "\t" that needed to be removed. Also, sometimes users wrote some URLs to link to personal content that had to be removed. At this stage, all Persian numbers were converted to English, and letters that had different alternatives were standardized to normalize the text. Then all the phrases were tokenized by defining the word boundary and converting the half-space to space. In the stemming stage, prefixes and suffixes used were removed.

After the pre-processing steps, the number of words in the Negative Feedback class was 6.1 million and the number of words in the Positive Feedback class was 34.1 million. Using the word cloud diagram, the most repetitive words in each of the classes can be depicted. Figures 6 and 7 show the repetitive words in the Positive Feedback and Negative Feedback classes, respectively. Words like "I gave back", "bad" and "I do not recommend" can be seen in the Negative Feedback figure, and words like "I'm satisfied", "Appropriate" and "Speed" can be seen in the Positive Feedback figure.

## Sentiment analysis

Data balancing is a crucial step that can increase accuracy. The random over-sampling method was used to balance the data. In other words, some data with the label of "Negative Feedback" were randomly selected and repeated. As a matter of fact, one of the common

**Figure 7 Negative feedback class word cloud.**

mistakes in this section is to apply the balancing method to the entire data which leads to errors in estimating the indicators. In these cases, the indicators are in a better position than the model capability and the results are reported incorrectly well. To avoid this, the balancing method was used only for the training data. After using the pseudo-labeling method, the number of positive feedbacks was about 1.8 million and the number of negative feedbacks was about 400 thousand. In this way, the negative feedbacks were repeated randomly about four times to balance the dataset.

The stratified *K*-fold cross-validation method is used to perform the evaluation. It is a method for model evaluation that determines how independent the results of statistical analysis on a dataset are from training data. In *K*-fold cross-validation, the dataset is subdivided into a *K* subset and each time one subset is used for validation and the other *K*-1 is used for training. This procedure is repeated *K* times and all data is used exactly once for validation. The average result of this *K* computing is selected as a final estimate. Stratified sampling is the process of dividing members of a dataset into similar subsets before sampling and this type of data sampling was selected due to imbalanced data. Using the stratified *K*-fold cross-validation method, we expect the values of the indicators to be more real. In all stages of measuring the accuracy of the model, *K* was considered equal to five.

As stated in the methodology, TF-IDF and fastText methods were used to extract the features. The BiLSTM and CNN models used the fastText output, and the Naïve Bayes and Logistics Regression models used the TF-IDF output, and their accuracy was finally compared with each other in Table 4. According to this table, the results of BiLSTM and CNN models are more accurate than others and CNN has given the best results.

As expected, due to the use of fastText and skip-gram methods, the need for data pre-processing has been reduced. In other words, stemming and normalization methods have not affected the final result. To examine this more closely, we chose the CNN model

**Table 4 Performance of different models based on AUC and F-measure.**

| States | Index | Fold1 | Fold2 | Fold 3 | Fold 4 | Fold 5 | Mean | Error (SEM) |
|---|---|---|---|---|---|---|---|---|
| BiLSTM | AUC: | 0.9934 | 0.9937 | 0.993 | 0.993 | 0.9934 | 0.9933 | $13.4 \times 10^{-5}$ |
| | F-score: | 0.9224 | 0.9244 | 0.9238 | 0.9216 | 0.9232 | 0.9230 | $49.6 \times 10^{-5}$ |
| CNN | AUC: | 0.9945 | 0.9945 | 0.9943 | 0.9946 | 0.9945 | 0.9944 | $4.9 \times 10^{-5}$ |
| | F-score: | 0.9293 | 0.9251 | 0.9306 | 0.9299 | 0.93 | 0.9289 | $99.2 \times 10^{-5}$ |
| Naïve Bayes | AUC: | 0.9877 | 0.9881 | 0.9878 | 0.988 | 0.9881 | 0.9879 | $8.12 \times 10^{-5}$ |
| | F-score: | 0.8856 | 0.8856 | 0.886 | 0.8863 | 0.8863 | 0.8859 | $15.7 \times 10^{-5}$ |
| Logistic Regression | AUC: | 0.9888 | 0.9891 | 0.9888 | 0.989 | 0.9881 | 0.9887 | $17.5 \times 10^{-5}$ |
| | F-score: | 0.8894 | 0.8901 | 0.8898 | 0.8895 | 0.8863 | 0.8890 | $69.1 \times 10^{-5}$ |

**Table 5 Performance of the CNN model in different situations based on AUC and F-measure.**

| States | Index | Fold1 | Fold2 | Fold 3 | Fold 4 | Fold 5 | Mean | Error (SEM) |
|---|---|---|---|---|---|---|---|---|
| Before Prep. | AUC: | 0.9943 | 0.9943 | 0.9944 | 0.9944 | 0.9945 | 0.994 | $3.7 \times 10^{-5}$ |
| | F-score: | 0.928 | 0.9298 | 0.9303 | 0.9271 | 0.9304 | 0.929 | $66.4 \times 10^{-5}$ |
| After Prep. | AUC: | 0.9945 | 0.9945 | 0.9943 | 0.9946 | 0.9945 | 0.994 | $4.8 \times 10^{-5}$ |
| | F-score: | 0.9293 | 0.9251 | 0.9306 | 0.9299 | 0.93 | 0.928 | $99.1 \times 10^{-5}$ |
| After Pseudo labeling | AUC: | 0.9944 | 0.9943 | 0.9946 | 0.9995 | 0.9996 | 0.996 | $12.5 \times 10^{-5}$ |
| | F-score: | 0.9431 | 0.9434 | 0.9443 | 0.9767 | 0.9758 | 0.956 | $80 \times 10^{-5}$ |

as the best model and we once performed the sentiment analysis process using the pre-processing steps and again without these steps. The AUC and *F*-score were 0.9943 and 0.9291 before pre-processing, and 0.9944 and 0.9288 after pre-processing. The results can be seen in Table 5. In the table, the meaning of the "before preprocessing" is just before the stemming and normalization steps. In other words, the methods used to create word embedding can depict the same words in the same range of spaces without the need to standardize letters and also without the need to identify the original root of words.

To implement pseudo-labeling, we developed a model that can estimate labels for unlabeled reviews using fastText and CNN models. After estimating all the labels, those with more than 90% probability for the Negative Feedback class and less than $1 \times 10^{-7}$ for the Positive Feedback class were selected. Therefore, 104.8 thousand Negative Feedback reviews and 30.5 thousand Positive Feedback reviews were labeled and could be used in the dataset for subsequent analysis. In using the pseudo-labeling technique, most of our focus was on Negative Feedback as a minority class, which also leads to balance the dataset as much as possible. In this way, a significant amount of unlabeled data that had been excluded from the sentiment analysis process was re-entered into the process and helped to increase the accuracy and generalizability of the model.

Contrariwise of pre-processing, the use of the pseudo-labeling method significantly improved the results. After using pseudo-labeling, the values of AUC and *F*-score improved to 0.996 and 0.956. The values of the three mentioned states can be seen based on different folds in Table 5. Figure 8 also shows the ROC curve for all three states.

The suggested model has had better results than the previous models which have used pre-processing methods in Persian sentiment analysis. For instance, some researchers

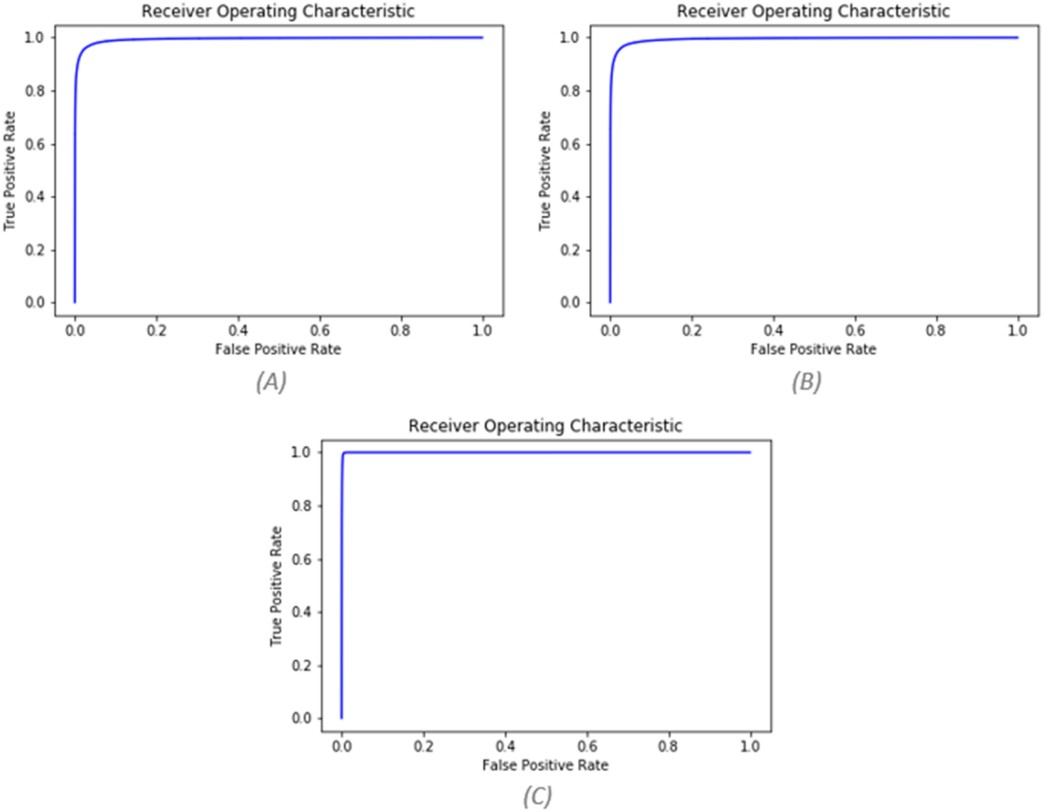

**Figure 8** **(A) AUC before pre-processing (AUC = 0.9943), (B) AUC after pre-processing (AUC = 0.9944), (C) AUC after pseudo labeling (AUC = 0.9996).**

introduced pre-processing algorithms and succeed to enhance the results of machine learning algorithms (*Saraee & Bagheri, 2013*). In the research, the *F*-score of the proposed pre-processing algorithms employing Naïve Bayes as a classifier algorithm is 0.878. In another research, the various alternatives for pre-processing and classifier algorithms were examined and the best result was assisted with an SVM classifier by 0.915 *F*-score value (*Asgarian, Kahani & Sharifi, 2018*). Also, some researches were attempted to utilize state-of-the-art deep models in such a way to reduce dependency on pre-processing and avoiding complex steps (*Roshanfekr, Khadivi & Rahmati, 2017*). The *F*-score of the BiLSTM and CNN algorithms in the research is 0.532 and 0.534. All mentioned article's focus was on the digital goods reviews in Persian two-class sentiment analysis as same as this article. A comparison of the results in this paper with other researches and other common algorithms indicates that not only has the dependence on data pre-processing been eliminated but also the accuracy has increased significantly.

The result reveals that it is quite possible to create independent models from the pre-processing process using the method of fastText and skip-gram. Moreover, BiLSTM and CNN methods can have significant results. However, all of the mentioned methods need to have an immense dataset. To prove this, It is noteworthy that the use of the pseudo-labeling method because of increasing training data has a great impact on the result. Independency from pre-processing steps is not related to the Persian language.

The results are reachable for other languages using sufficient labeled samples and the mentioned methods in this article.

## CONCLUSION

The dataset included approximately 3 million reviews was extracted from the digital goods section of the Digikala website as the largest online store in Iran. Basic pre-processing methods were used to modify the words and tokenize them. Due to the lack of labels for a large part of the dataset, the pseudo-labeling method was employed which improved the results. Data balancing was also performed using random over-sampling. Persian data pre-processing was found difficult, so the fastText method was conducted to reduce the need for data pre-processing and word embedding development. The embeddings were employed as the input to the BiLSTM, and CNN models. Using the suggested model, not only have the obtained results been very desirable and are much more accurate in Persian compared to other reports but also there are no complications related to data pre-processing. The effect of stemming and normalization on the output was evaluated and revealed that the proposed method is not dependent on data pre-processing. Eventually, Besides the comparison of machine learning and deep learning methods in sentiment analysis, the TF-IDF and fastText methods were compared to create word embedding. The best result was associated with fastText and CNN. The main achievement of this model is the reduction of the need for data pre-processing. Data pre-processing in English is convenient and accurate due to the expanded text pre-processing libraries. However, in other languages, data pre-processing is very complicated because of the lack of proper libraries. Over the suggested model was proved that this need is largely solvable (AUC = 0.996) and the pre-processing steps can be reduced to preliminary tokenization processes. Avoiding complex text preprocessing is also important for other languages Since most text preprocessing algorithms have been developed for English and cannot be used for other languages. In other words, the taken steps are possible to be implemented for other languages to achieve the same results independently of pre-processing steps. Moreover, the created word embedding due to its high accuracy can be used in other text analysis problems especially related to online digital goods.

### Funding
The authors received no funding for this work.

### Competing Interests
The authors declare that they have no competing interests.

### Author Contributions

- Sajjad Shumaly conceived and designed the experiments, performed the experiments, analyzed the data, performed the computation work, prepared figures and/or tables, authored or reviewed drafts of the paper, and approved the final draft.

- Mohsen Yazdinejad conceived and designed the experiments, performed the experiments, analyzed the data, performed the computation work, prepared figures and/ or tables, and approved the final draft.
- Yanhui Guo conceived and designed the experiments, authored or reviewed drafts of the paper, and approved the final draft.

## Data Availability

Data and code are available at GitHub repository:

https://github.com/mosiomohsen/persian-sentiment-analysis-using-fastText-word-embedding-and-pseudo-labeling.

## Supplemental Information

Supplemental information for this article can be found online at http://dx.doi.org/10.7717/peerj-cs.422#supplemental-information.

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
