# Peer review of "Persian sentiment analysis of an online store independent of pre-processing using convolutional neural network with fastText embeddings"

_PeerJ Computer Science, doi:10.7717/peerj-cs.422_

## Round 0.1 · original submission · Major Revisions

Improve the description of the method while emphasizing its novelty. Present more details of the experiments. Compare your method and results with the state-of-the-art.

Reviewer 1 ·

Basic reporting

The article lacks the scientific soundness.
The structure, tables and figures must be modified (complemented, removed, added references into their source).

Experimental design

Methods are not clearly presented (pre-processing methods, fastText vectorization methods).
The only method is tested: no investigation on different classifiers (different their architectures, hyper-parameter values) or vectorization types (with different parameters), etc.

Validity of the findings

The novelty is questionable.
Too little information about the used dataset.
Having no details on the used dataset, used pre-processing methods, used fastText embeddings, it is hard to interpret the results.

Additional comments

There are my remarks:
1. The main problem here: the contribution of your work is not clear. You have solved the sentiment analysis problem with only one classifier (CNN) offered by Kim (not testing any other architectures or hyper-parameter values). The classifier was applied on the fastText embeddings (classic neural vectorization technique, that recently is more and more replaced with more sophisticated transformer methods as, e.g., BERT).
2. You emphasize the importance of pre-processing (Section 2.2), but provide no details on how pre-processing was performed in their experiments. Please, write what normalization, tokenization, stemming tools have you used or implemented yourself. Please, provide details on how accurate they are and how much impact the sentiment analysis task, etc.
3. The authorship of Figure 1 is not yours, please, put the reference.
4. Figure 2 is not very informative: please, provide the exact details on each block.
5. How categories in Figure 3 make the impact on your sentiment analysis task? Please, provide the statistics about the used sentiment dataset. How many instances were in the positive and negative classes after balancing? And if you remove some instances from the original dataset does Figure 3 still make sense?
6. Table 1, Figure 4, Figure 5 are not clear. You either must provide translations or remove table and figures from your paper completely.
7. The obtained very high f-score values (over 0.9) seem suspicious. Maybe your positive/negative splits had too little diversity within each class? It even makes the impression that training/evaluation was performed on the imbalanced dataset. Would you get similar results if you would randomly select other instances from the major class?
8. Without having the exact details about the used dataset, I cannot interpret the results and must accept your conclusions with caution.

Reviewer 2 ·

Basic reporting

1. The use of English language is not proper.
2. Related work is not appropriate or missing.

Experimental design

1. Experimental design is not suitable to evaluate the efficacy of the proposed approach.

Validity of the findings

Results are not evaluated properly. Further experimentation is required.

Additional comments

1. ‘Related work’ is missing. Split the ‘Introduction’ into introduction and ‘Related Work’ for completeness.
2. Line 94. Authors assume that the use of fastText would generate better results. In research, such assumptions are not appropriate. So either use multiple algorithms for this task to evaluate their efficacy or support your assumption with the findings from other research works.
3. The fastText model uses the skip-gram models works in similar fashion to CBOW so why authors prefere fastText over CBOW, TF/IDF, GLoVe, etc. I strongly recommend the authors to use CBOW, TF/IDF and GLoVe and compare their performance with fastText.
4. The contributions of the study are not highlighted properly. At the end of Introduction, describe the contribuions in a bulleted form for readability.
5. The choice of using CNN is not justified. LSTM has proven to show better performance for machine learning tasks.
6. The performance of the proposed approach is not evaluated. The best way to do that is to compare the performance of the proposed approach with SVM, logistic regression, GBM, and at least two state-of-the-art approaches.
7. The problem of two class classification is simple. I suspect the performance of the proposed approach will degrade if more classes are added. So I ask the authors to perform classification on ‘positive’,’neutral’, and ‘negative’ grounds.
8. English grammer and typos.
a. Always use Sentence case for writing. Many sentences in the manuscript start with small alphaets.
b. When using numerical values in the text, use them properly, e.g., Section 3.3 ‘’10-7’ to ’10-7’.
c. Extensive English proof reading and correction is required by a native speaker.

---

## Round 0.2 · Minor Revisions

Extend the discussion. The authors should discuss the applicability of the proposed methods for other languages as well as the implications of this study for future researcher efforts in sentiment analysis in various languages.

Reviewer 1 ·

Basic reporting

no comment

Experimental design

no comment

Validity of the findings

no comment

Additional comments

Thank you for considering my previous comments.
My main concern is still about a focus on one specific language. Maybe you could add some recommendations for other languages on how they can benefit from your research.

Reviewer 2 ·

Basic reporting

No comment

Experimental design

No comment

Validity of the findings

No comment

Additional comments

Authors resolved all the issues.

---

## Round 0.3 · accepted · Accept

The article is accepted for publication.